# Antrodan Alleviates High-Fat and High-Fructose Diet-Induced Fatty Liver Disease in C57BL/6 Mice Model via AMPK/Sirt1/SREBP-1c/PPARγ Pathway

**DOI:** 10.3390/ijms21010360

**Published:** 2020-01-06

**Authors:** Charng-Cherng Chyau, Hsueh-Fang Wang, Wen-Juan Zhang, Chin-Chu Chen, Shiau-Huei Huang, Chun-Chao Chang, Robert Y. Peng

**Affiliations:** 1Research Institute of Biotechnology, Hungkuang University, No. 1018, Sec. 6, Taiwan Boulevard, Shalu District, Taichung City 43302, Taiwan; n404user@gmail.com; 2Institute of Biomedical Nutrition, Hungkuang University, No. 1018, Sec. 6, Taiwan Boulevard, Shalu District, Taichung City 43302, Taiwan; fang54@hk.edu.tw (H.-F.W.); yuchin831112@gmail.com (W.-J.Z.); 3Grape King Biotechnology Center, 60, Sec 3, Longgang Rd., Chung-Li City, Taoyuan County 320, Taiwan; gkbioeng@grapeking.com.tw; 4Division of Gastroenterology and Hepatology, Department of Internal Medicine, Taipei Medical University Hospital, Taipei 11301, Taiwan; 5Division of Gastroenterology and Hepatology, Department of Internal Medicine, School of Medicine, College of Medicine, Taipei Medical University, Taipei 11301, Taiwan; 6Research Institute of Medical Sciences, School of Medicine, Taipei Medical University, Taipei 11301, Taiwan; 7School of Medicine and Health, Hungkuang University, No. 1018, Sec. 6, Taiwan Boulevard, Shalu District, Taichung City 43302, Taiwan

**Keywords:** *Antrodia cinnamomea*, Antrodan, orlistat, high-fat-high-fructose diet, non-alcoholic fatty liver disease (NAFLD), insulin, hepatoprotective

## Abstract

Non-alcoholic fatty liver disease (NAFLD) and -steatohepatitis (NASH) imply a state of excessive fat built-up in livers with/or without inflammation and have led to serious medical concerns in recent years. Antrodan (Ant), a purified β-glucan from *A*. *cinnamomea* has been shown to exhibit tremendous bioactivity, including hepatoprotective, antihyperlipidemic, antiliver cancer, and anti-inflammatory effects. Considering the already well-known alleviating bioactivity of *A. cinnamomea* for the alcoholic steatohepatitis (ASH), we propose that Ant can be beneficial to NAFLD, and that the AMPK/Sirt1/PPARγ/SREBP-1c pathways may be involved in such alleviations. To uncover this, we carried out this study with 60 male C57BL/6 mice fed high-fat high-fructose diet (HFD) for 60 days, in order to induce NAFLD/NASH. Mice were then grouped and treated (by oral administration) as: G1: control; G2: HFD (HFD control); G3: Ant, 40 mgkg (Ant control); G4: HFD+Orlistat (10 mg/kg) (as Orlistat control); G5: HFD+Ant L (20 mg/kg); and G6: HFD+Ant H (40 mg/kg) for 45 days. The results indicated Ant at 40 mg/kg effectively suppressed the plasma levels of malondialdehyde, total cholesterol, triglycerides, GOT, GPT, uric acid, glucose, and insulin; upregulated leptin, adiponectin, pAMPK, Sirt1, and down-regulated PPARγ and SREBP-1c. Conclusively, Ant effectively alleviates NAFLD via AMPK/Sirt1/CREBP-1c/PPARγ pathway.

## 1. Introduction

Patient with non-alcoholic fatty liver disease (NAFLD) implies a state of excessive fat built-up in livers with, or without minimal inflammation [1,2,3,4]. NAFLD is the hepatic manifestation of the metabolic syndrome associated with obesity [5]. T2DM, if associated with NAFLD, would become very complicated and makes diabetes management more challenging [6]. DM appears to promote the development of NAFLD and increases the risk of cirrhosis and hepatocellular carcinoma [6].

A high-fat diet enhances intestinal permeability by modulating the intestinal tight junctions, secretion of the barrier-disrupting hydrophobic bile acids, pro-inflammatory signaling cascades, oxidative stress, and apoptosis of intestinal epithelial cells, as well as alters the barrier-disrupting gut microflora [6]. On the other hand, high fructose consumption (HFC) leads to increased body weight with elevated systolic blood pressure, blood glucose, insulin, and serum triglyceride (TG) levels [7]. HFC reduces energy expenditure, thereby causing obesity, adipocyte hypertrophy, and inflammation [8]; lipid spillover further causes hepatic steatosis, peripheral insulin resistance and diabetes, raised levels of LDL and a decrease in HDL [9]. Furthermore, HFC elicits elevation of certain pro-inflammatory serum proteins [10]. Worth noting, a high-fructose and high-fat diet potentially tends to damage liver mitochondria, increasing the risk from fatty-liver disease and metabolic syndrome [8].

Much of the literature has implicated the beneficial bioactivities of *A*. *cinnamomea,* including anti-adenocarcinoma, antihypertension, antileukemia, antiliver cancer, anti-inflammation, hepato-protection against CCl_4_– and ethanol–induced liver injuries [11,12]. Previously, we showed that the extract of *A. cinnamomea* alleviated the bladder transitional cell carcinomas (TCC) [12], and showed a potent anti-metastatic effect via inhibiting the matrix metalloproteinase (MMP) -2 and -9.activities [13]. However, there has been little research into understanding how bioactive polysaccharide of *A. cinnamomea* affects the fatty liver diseases. 

The mycelia of *A. cinnamomea* contains polysaccharides 16.97%, from which five fractions of polysaccharides were isolated and denoted as fractions AC-1 to AC-5 [14]. Antrodan, a β-glucan obtained by further treatment of the AC-2 fraction, was named as “Antrodan” [14]. Fraction AC-2 demonstrated a rather potent anti-inflammatory capability [15], while astonishingly, we recognized that Antrodan exhibited potent heptoprotective [16], as well as anti-benign prostate hyperplasia (BPH) [17]. On the other hand, Antrodan prevented the epithelial-mesenchymal transition (EMT) and exhibited promising anti-inflammatory hypolipidemic bioactivities [17]. Antrodan was found beneficial for alleviating lung cancer [18] and antimetastatic effects [13]. 

As widely recognized, AMP-activated protein kinase (AMPK) pathway is a master cellular energy metabolic switch relevantly associated with positive lipid regulation in the liver; and AMPK is well-established as the therapeutic target of NAFLD [19,20]. On the other hand, among seven mammalian sirtuins (silent information regulator T, SIRTs), Sirt1 1 is the most extensively studied, due to its many positive functions in both AFLD and NAFLD [21]. Both pAMPK and Sirt1 synergistically suppressed the expression of PPARγ, leading to the inhibited lipid synthesis [21].

Considering the already well-known alleviating bioactivity of *A. cinnamomea* for the alcoholic steatohepatitis (ASH) [22], and up to the present, there is no licensed drug that has been clinically approved for the treatment of NAFLD [23]. Therefore, we propose that Antrodan can be beneficial to the NAFLD and that the AMPK/Sirt1/PPARγ/SREBP-1c pathways may be involved in the alleviation of NAFLD by Antrodan. To uncover this, a framework shown in Figure 1 was conducted to carry out a mice-model fed on the high fat and high fructose diet to induce NAFLD, and examine the alleviative effects of Antrodan on these NAFLD-affiliated mice.

## 2. Results

### 2.1. The Retarding Effect of Antrodan Against the HFD Regarding the Liver- and Body-Weight 

HFD significantly increased the body- and liver-weights of mice. The body- and liver-weights and the ratio liver wt/body wt in the Antrogen (40 mg/kg) control group remained normal as the control (Table 1). Expectedly, a high dose Antrodan cotreatment in HFD significantly suppressed the body- and liver-weights, and the ratio liver wt/body wt, being more effective than the positive control ‘Orlistat’ (Table 1).

### 2.2. Effect of Antrodan on Plasma Levels of Malondialdehyde, Total Cholesterol, Triglyceride, and Ratio LDL-C/HDL-C 

The plasma levels of malondialdehyde were highly stimulated by HFD (*p* < 0.001) (Figure 2A). C fr1otreatment with Antrodan significantly reduced the lipid peroxidation (*p* < 0.01, low dose Antrodan, 20 mg/kg; and *p* < 0.05, the high dose Antrodan, 40 mg/kg) (Figure 2A). Similar trends were observed for the plasma levels of total cholesterol (Figure 2B) (*p* < 0.001) and triglyceride (Figure 2C) (*p* < 0.001). Antrodan alone could not affect the plasma total cholesterol levels, but efficiently suppressed the plasma total cholesterol levels when co-treated with HFD, despite high or low doses, and it seemed that its effect was comparable with Orlistat (*p* < 0.001) (Figure 2B). In contrast, Antrodan (40 mg/kg) seemed to have significantly suppressed the plasma triglyceride levels, compared to the control, and as a consequence, when co-treated with HFD, the plasma triglyceride levels were significantly reduced (*p* < 0.001) (Figure 2C). HFD significantly increased the LDL-C/HDL-C ratio (*p* < 0.001), unexpectedly, Antrodan (40 mg/kg) showed a similar effect (*p* < 0.01) (Figure 2D). In all other HDF-co-treated groups all the ratios were highly raised (*p* < 0.001).

### 2.3. Effect of Antrodan on the Plasma Levels of Glucose, Insulin, Leptin and Adiponectin

When compared to the control, HFD significantly stimulated the plasma levels of glucose and leptin (*p* < 0.001), moderately elevated insulin, and reduced adiponectin levels (Figure 3), and Antrodan alone (40 mg/kg) effectively suppressed the plasma levels of glucose, leptin, insulin, and adiponectin (Figure 3). When co-treated with HFD, Orlistat showed very promising suppressing effect on the plasma glucose level, compared to the low Antrodan (20 mg/kg) (*p* < 0.001) (Figure 3). Strangely, high doses Antrodan (40 mg/kg) were less effective (*p* < 0.05). The insulin level was significantly suppressed by high dose Antrodan (40 mg/kg), as well, in the co-treated groups. Antrodan was ineffective for suppressing the elevated leptin level when compared to the HFD group (Figure 3), but slightly effective to raise the level of adiponectin.

### 2.4. Effect of Antrodan on the Activities of Plasma Levels of GOT, GPT, and Uric Acid

The plasma levels of GOT, GPT, and uric acid were all significantly elevated by HFD (*p* < 0.001) (Figure 4). Antrodan (40 mg/kg) was safe and did not show any harmful damage to the liver. In combined treatments, Orlistat, Antorodan, despite low or high dose all showed significantly reduced levels of GOT (*p* < 0.001) and GPT (*p* < 0.05). As for uric acid, Orlistat (*p* < 0.01) was seen more effective than Antrodan (*p* < 0.05) (Figure 4).

### 2.5. Histopathological Findings

According to the staging system for steatohepatitis [24], the results shown in hematoxylin-eosin (H&E) stain (Figure 5) revealed that the liver tissues were damaged with severe steatohepatitis when induced by HFD (Figure 5b), compared to the control (Figure 5a) and that of Antrodan (40 mg/kg) treated (Figure 5c). A vast number of large fat droplets accumulated in the liver tissues with inflammation. Abundant number of inflammatory cells, in particular eosins, also appeared in the inflammation sites. Similar results were previously demonstrated by Wang et al. [25], who indicated that high-fat diet-induced hepatocellular steatosis, ballooning degeneration, lobular inflammation, spotty focal necrosis that were gradually shown in the hepatic lobule, especially in zone 3 of acinus. In the study, at the end of HFD treatment (day 105), steatohepatitis established with inflammatory cell infiltration and spotty focal necrosis. Compared to Orlistat (Figure 5d), Antrodan partially alleviated these pathological changes after being treated for 45 days (Figure 5e,f).

### 2.6. Protein Expressions Affected by Antrodan

HFD inhibited the expression of Sit1 compared to the control (*p* < 0.01) (Figure 6). Antrodan alone did not show any effect. In the co-treated groups, Orlistat, Antrodan at 20 mg/kg, and 40 mg/kg significantly upregulated Sirt1 at significant levels of *p* < 0.001, *p* < 0.01, and *p* < 0.001, respectively (Figure 6). The levels of AMPK and p-AMPK were downregulated by HFD (*p* < 0.001 and *p* < 0.05, respectively), was significantly upregulated by co-treatment with Orlistat (*p* < 0.05), Antrodan (20 mg/kg) (*p* < 0.001), and Antrodan (40 mg/kg) (*p* < 0.05) (Figure 6).

The level of PPARγ was highly upregulated (*p* < 0.05) by HFD, while Antrodan (40 mg/kg) alone significantly downregulated (*p* < 0.001) its level (Figure 7). When Antrodan was cotreated with HFD, the level of PPARγ was found significantly suppressed by Orlistat (*p* < 0.01), Antrodan (20 mg/kg) (*p* < 0.05), and Antrodan 40 mg/kg (*p* < 0.001), respectively (Figure 7). As contrast, the level of SREBP-1C was highly elevated by HFD (*p* < 0.001). Antrodan (40 mg/kg) alone did not show any effect. Among the cotreated groups, Orlistat, Antrodan (20 mg/kg), and Antrodan (40 mg/kg) revealed to be significantly effective regarding the suppression of SREBP-1C (each *p* < 0.05) (Figure 7).

## 3. Discussion

### 3.1. The Adverse Metabolic Role of a High Fat Diet

High fat diet induces oxidative stress and apoptosis in intestinal epithelial cells [6]. In addition, a high-fat diet enhances intestinal permeability directly by stimulating pro-inflammatory signaling cascades and indirectly via increasing barrier-disrupting cytokines like TNFα, interleukin (IL) 1B, IL6, and interferon γ (IFNγ), and decreasing barrier-forming cytokines like IL10, IL17, and IL22 [6].

### 3.2. Fructose Pays a Higher Energy Cost Regarding the ATP Production

The consumed fructose is first converted into glucose in the liver, then followed by glucose oxidation in the extrahepatic cells, this reaction pathway requires the use of an additional 2 ATPs compared to that for the direct oxidation of blood glucose, which means this is associated with a higher ATP used/ATP synthesized ratio, and thus, a higher energy cost of net ATP gained [26]. The diet-induced thermogenesis (DIT) is always higher for fructose than glucose [26]. Biochemically, for glucose metabolism, it can be estimated that 2 moles of ATP are used and 26.5 moles ATP are synthesized, corresponding to a net gain of 27.5 moles ATP/mole glucose. Since the initial energy content of one mole glucose is 686 kcal, the energy efficiency of glucose oxidation, i.e., the energy cost of ATP gained, can be estimated as 686/27.5, or 24.9 kcal/mole ATP [26]. In contrast, when fructose is oxidized as lactate in extrahepatic cells, the overall number of ATP used (2 ATP) and synthesized (26.5 ATP) is the same as for glucose oxidation, and the overall energy efficiency is therefore similar to that of glucose. However, 2 ATP are used in the liver, while 26.5 ATP are synthesized in extrahepatic cells. As a consequence, the energy cost of ATP gained increases to 26.9 kcal/mole. This corresponds to an 8% increase compared to glucose [26]. 

### 3.3. How Does the High Fructose Diet Affect Llipid Metabolism?

After ingestion, the fructose molecules can be rapidly absorbed through the glucose transporter-5 (GLUT5) and released into the bloodstream. Fructose is then absorbed mainly by the liver cells that exhibit high amounts of GLUT2 [27] (Figure 8). In contrast, virtually no fructose can be absorbed by pancreatic beta cells due to extremely low affinity of the pancreatic beta cell GLUT2 and GLUT5 transporters for fructose (https://www.ncbi.nlm.nih.gov/gene/6514). On the other hand, glucose can trigger the release of insulin from pancreatic beta cells, but fructose is unable to stimulate insulin secretion [28]. High fructose consumption leads to the accumulation of adipose tissue, systemic inflammation, obesity, oxidative stress, and consequently insulin resistance in different tissues [9,25,29,30].

Mechanistically, fructose-1-P (F-1-P) exhibits multiple effects to initiate fatty liver. F-1-P over upregulates cytoplasmic malonyl-coA, inhibiting the carnitine palmitoyl transferase 1 (CPT-1), thereby, retarding the transport of lipids into mitochondria and subsequent β-oxidation [31]. Simultaneously, F-1-P activates peroxisome proliferator-activated receptor-gamma coactivator 1 beta (PGC-1β) protein, which in turn increases the expression of the sterol regulatory element-binding protein 1c (SREBP1c). SREBP1c initiates the transcription of fatty acyl-CoA synthase (FAS) and acetyl-CoA carboxylase (ACC) proteins [32].

The over-produced fatty acids in cells now have three different ways to dispose: (1) Part of the triglycerides deposit in the hepatocytes, leading to NAFLD. (2) Another part binds to apolipoprotein (ApoB) to produce VLDL; or (3) Part of them simply diffuses in form of free fatty acids into the bloodstream, causing hypercholesterolaemia and dyslipidemia [33].

### 3.4. Why is Antrodan Ineffective at Suppressing the Ratio LDL-C/HDL-C? 

NAFLD is the liver injury most often associated with disordered insulin resistance, including obesity, diabetes, and the metabolic syndrome [34]. To induce such a complicated pathological syndrome, a simple high-fat diet would be impossible. High-fat and high-fructose diets have been used to induce animal model diabetes mellitus to evaluate the effect on change of leptin level [35]. Studies indicate that fructose may be a carbohydrate with greater obesogenic potential than other sugars [8,36]. Fructose promotes and complicates glucose metabolism, enhancing the accumulation of triacylglycerol in the hepatocytes, and causing alterations in the lipid profile associated with severe inflammatory responses, simultaneously stimulating huge production of ROS leading to a systemic etiology with insulin resistance [8]. A high-fructose diet can cause hyperinsulinemia, while a high-fat diet can result in impaired pancreatic function of insulin secretion and glucose intolerance, indicating that a high-fructose diet and a high-fat diet may exert divergent effects on glucose metabolism in rats [35]. As can be imagined, such a highly complicated syndrome, in fact, would not be simply alleviated by a single medicine. Antrodan alleviated MDA, TC, and TG levels, but failed to suppress the ratio LDL-C/HDL-C (Figure 2). Orlistat inhibits lipases in the gastrointestinal tract, preventing the absorption of approximately 30% of dietary fat [37], revealing a non-systemic treatment for obesity. Although, the literature indicates that Orlistat improves lipid profiles in non-diabetic obese patients, reducing levels of total cholesterol, and low-density lipoprotein cholesterol [37]. However, Orlistat also failed to rescue the LDL-C/HDL-C ratio (Figure 2).

The reasons can be attributed to the increased LDL-C formation when induced by HFD. The studies of Fernandez and West have demonstrated that, among the saturated fatty acids (SFAs), stearic acid (18:0) appears to have a neutral effect on LDL-C, while lauric (12:0), myristic (14:0), and palmitic (16:0) acids are considered to be hypercholesterolemic. SFAs increase plasma LDL-C by increasing the formation of LDL in the plasma compartment and by decreasing LDL turnover [38].

### 3.5. Animal Model Selection Affects the Experimental Outcomes

In addition, the selection of the animal model will also greatly affect the outcome of experiment. Compared to the dyslipidemic subject who exhibits LDL-C = 154 ± 7; HDL-C= 48 ± 4 mg/dL; and LDL-C/HDL-C = 3.21), the corresponding values of C57BL/6 mice are: LDL-C = 21 ± 2; HDL-C = 97 ± 4 mg/dL; and LDL-C/HDL-C = 0.26 [39]. Obviously, the wide genetic variation exists, implying that animal models, in fact, can only be used as a clinical reference. A major difference of mouse models from humans is that mice lack cholesteryl ester transport protein (CETP) [40]. CETP is the key enzyme involved in plasma cholesterol transport, which transfers cholesteryl ester (CE) from HDL to apoB-containing lipoproteins such as LDL and VLDL [40]. Rats, dogs, and pigs also have no- or low-plasma CETP activities, and they all display high high-density lipoprotein cholesterol (HDL-c) and low LDL-c plasma lipoprotein distribution, similar to mice, which is associated with a low risk of CVD [40].

### 3.6. Antrodan treatments Appear to be Effective in Regulating Adiponectin but not in Leptin Levels 

Fructose affects the sensation and response of the central nervous system via elevation in cannabinoid 1 (CB1) receptor messenger RNA (mRNA) and leptin [35], hence, it might disturb hunger and satiety control, as well as contribute to the development of obesity and metabolic complications [8].

Increased circulating levels of leptin (a proinflammatory adipokines) in obesity lead to hypothalamic leptin resistance, turning down anorexigenic and energy expenditure signals and further contribute to aggravate obesity [41]. Increased levels of pro-inflammatory adipokines (e.g., leptin), and decreased levels of anti-inflammatory adipokines (e.g., adiponectin) in obesity may produce a chronic state of low-grade inflammation and promote the development of insulin resistance and type-2 diabetes, hypertension, atherosclerosis and other cardiovascular diseases, and some types of cancer [42]. Moreover, since adiponectin also acts as an insulin-sensitizing hormone in muscles and the liver, lower levels of adiponectin further contribute to peripheral insulin resistance in obesity [22]. Although highly elevated leptin level seemed to be not affected from the Antodan treatment, the slightly raised adiponectin level (Figure 3) was found to be sufficiently high enough to activate AMPK into pAMPK (Figure 6).

### 3.7. Elevated PPARγ and SREBP-1c Increased Lipid Synthesis

The hepatic PPARγ plays a putative role in the progression of fatty liver disease in the NAFLD patients [43]. Hepatic PPARγ and the nuclear hormone receptors liver X receptor α (LXRα) independently regulate lipid accumulation in mice livers [21]. Signals of LXRα-SREBP-1c and ChREBP upregulate the expression of lipogenic genes in both normal and obesity mice livers [44]. Normally, the PPARγ expression in normal mice livers is low, but can be highly upregulated in fatty livers [45]. Similar evidence also indicated that PPAR-γ is up-regulated in the liver of obese patients with NAFLD, and recently, the expression of PPAR-γ is considered as an additional reinforcing lipogenic signal, assisting SREBP-1c to trigger the development of hepatic steatosis [43]. The literature has indicated that, in fact, PPARγ could be upregulated as early as two weeks after being induced with high fat diet [46]. We showed hepatic PPARγ was significantly upregulated (*p*< 0.05) (Figure 7) in mice liver tissues associated with a huge amount of lipid deposit after 60 day-induction with HFD (Figure 5b). The hepatic PPARγ signal is highly expressed in the fatty livers, compared to normal mice, implicating the fact that, hepatic PPARγ may contribute more significantly to the development of fatty liver than LXRα, consistent with our findings (Figure 7). Both pAMPK and Sirt1 synergistically suppressed the expression of PPARγ, leading to the inhibited lipid synthesis [21].

### 3.8. Sirt1 and pAMPK Inhibited PPARγ and SREBP-1c, thereby, Suppressed Lipid Synthesis and Alleviated Insulin Resistance

Increased levels of pro-inflammatory adipokines (e.g., leptin) (Figure 3), and decreased levels of anti-inflammatory adipokines (e.g., adiponectin) (Figure 3), in obesity may produce a chronic state of low-grade inflammation and promote the development of insulin resistance and type-2 diabetes, hypertension, atherosclerosis, and other cardiovascular diseases, as well as some types of cancer [42].

AMP-activated protein kinase (AMPK) and the histone/protein deacetylase Sirt1 are fuel-sensing molecules, that regulate each other and share many common target molecules [47]. AMPK is a fuel-sensing enzyme that is activated by decreases in AMP/ATP ratio in the cells [48,49]. Sirt1 is widely expressed in mammalian cells and has been studied in many tissues, including liver, skeletal muscle, adipose tissue, pancreas (β-cells), brain [50], and the endothelium [51].

Among the seven mammalian homologs of sirtuin (Sirt1-7) [52], Sirt1 is the most extensively studied member, and is involved in both NAFLD and AFLD [39,53]. Like AMPK, Sirt1 responds to increases and decreases in nutrient availability (caloric restriction or starvation) [54], energy expenditure [55], and antioxidant mechanism [56]. pAMPK changes the NAD^+^ abundance and the NAD^+^/NADH ratio to upregulate Sirt1 [57], which in turn plays beneficial roles in modulating hepatic lipid metabolism, hepatic oxidative stress, and mediating hepatic inflammation through deacetylating some transcriptional regulators against the progression of fatty liver diseases [20]. In recent years, the evidence has suggested that SIRTs play important roles in regulating the fatty liver disease-related metabolic processes [20]. Antrodan elevated the level of adiponectin (Figure 3) and suppressed that of PPARγ and SREBP-1c, thereby inhibiting the lipid biosynthesis and promoting obese- and steatohepatitis-associated insulin resistance. Hence, the levels of GPT, GPT, and uric acid (Figure 4) were all improved. Worth noting, mice livers have the highest NAD^+^/NADH ratio (4.0) compared to the liver of swine (0.07) [58], implicating the feasibility of this mice model.

To summarize, the pathways whereby Antrodan alleviated the NAFLD induced by HFD, as summarized in Figure 9. Biochemically, fructose associated with free fatty acids (FFA) played many adverse roles, inducing NAFLD. FFA activated the production of long chain Acyl CoA, while fructose enhanced the expression of SREBP-1c, activating FAS and ACC1, and increasing the synthesis of malonyl CoA, the latter inhibits CPT-1 retarding the transport of extramitochondrial LCAcCoA into mitochondrial outer membrane, and thereby, reduced the β-oxidation. On the other hand, fructose produced a vast amount of glucose, inducing insulin resistance and ATP depletion, resulting in increased low energy index ‘AMP/ATP’, which, together with the adiponectin, raised by Antrodan stimulated AMPK phophorylation, the increased pAMPK in turn stimulated the NAD^+^/NADH ratio, and induced expression of Sirt1. Sirt1 activated mitochondrial biogenesis, and together with pAMPK inhibited PPARγ/SCREBP-1c induced FAS activity and TG levels to alleviate the NAFLD (Figure 9). Thus, it is evidently seen that Antrodan improved NAFLD via the AMPK/PPARγ/ SCREBP-1c pathway.

## 4. Materials and Methods

### 4.1. Chemicals and Antibodies

Antibodies against AMPK (ab131512), p-AMPK (ab23875), PPARγ (ab45036) and SREBP-1c (ab26481), and HRP (horseradish peroxidase)-conjugated goat anti-(rabbit IgG) antibody (ab97051) were purchased from Abcam (Cambridge, UK). Sirt1 (131611AP) antibody was provided by Proteintech (Rosemont, IL, USA). β-Actin (tcba13655) was a product from Taiclone (Taipei, Taiwan). The high-fat and high-fructose diet (HFD) (Research Diet D17010102) was supplied by the Research Diets, Inc. (New Brunswick, NJ, USA), which contains 40% fat, 22% fructose, and 2% cholesterol. The regular or standard diet (3.3 kcal/g) contained 58.9% carbohydrate, 26.7% protein, and 12.4% crude fat. Orlistat was purchased from Sigma-Aldrich (St. Louis, MO, USA).

### 4.2. Source of Antrodan

Antrodan from the *Antrodia cinnamomea* mycelia was prepared as previously reported [14] with slight modification. In brief, the lyophilized defatted mycelia (1 kg) were suspended in water (1:10, *w*/*v*) and heated at 80 °C for 2 h to remove the water soluble substances. The residue was extracted with hot alkaline solution (pH 9.0, 1:10 *w*/*v*) at 80 °C for 2 h and filtered. The residue was repeatedly extracted for additional two times. The three extracts were combined. Afterwards, the pH value was adjusted to 4.0 using 1 N HCl solution, the solution was left to stand at 4 °C overnight to facilitate the precipitation. The precipitate was collected by centrifuging at 3500× *g* for 30 min and subjected to dialysis against the deionized water (DDW) for 3 days to remove the free sugars and amino acids (dialysis tube MW cut-off 12,000–16,000 Da, Wako, Japan), then lyophilized to obtain base-soluble extract. The extract containing Antrodan was loaded onto a Sepharose CL-6B column (3.0 × 82 cm) and eluted with DDW (pH 11.0 adjusted with 1N NaOH) at a flow rate 0.5 mL/min to separate the polysaccharides and collect the target Antroden with a fraction collector. The yield was 9.19% (*w*/*w*) with an average molecular weight of 442 kDa, as analyzed by the high-performance size-exclusion chromatography (HPSEC).

### 4.3. Induction of Fatty Liver Diseases and Treatment with Antrodan

In the previous study [59], the non-genetically modified C57BL/6 mice when exposed to the high-fat high-carbohydrate (HFHC) diets may result in increased body weight, body fat mass, fasting glucose, and insulin-resistant level, compared with chow mice. Based on the study, we applied the high-fat and high-fructose diet (HFD) to conduct the hypothesis that mice given ad libitum access to the HFD would induce increased hepatic lipid accumulation and generate significant fatty liver symptoms. Sixty C57BL/6 male mice, 6-week-old, were provided by the BioLASCO Taiwan Co., Ltd. (Taipei, Taiwan). This experiment was carried out according to the regulation controlled by the Animal Research Committee of Hungkuang University (HKU), and all experimental protocols were approved by the Ethical Committee of HKU (approved affidavit No. HK-P-10613). The mice were housed in a pathogen-free room under controlled temperature (25 ± 2 °C), relative humidity (65 ± 5%), and alternating 12h-light/12h-dark cycles. For the first week, the mice were acclimated, then randomly divided into 6 groups (*n* = 10 mice in each group) as follows: Group 1, the blank control fed standard regular diet; Group 2, fed high-fat and high-fructose diet (HFD); Group 3, the Antrodan positive control fed Antrodan 40 mg/kg; Group 4, fed combined HFD with Orlistat (10 mg/kg); Groups 5 and 6, fed HFD, which was supplemented with Antrodan 20 mg/kg (low dose), and 40 mg/kg (high dose), respectively. For induction, the mice were first fed HFD for 60 days, and then fed on HFD and tube irrigated with Antrodan or Orlistat, once daily for further 45 days (Figure 1). Antrodan or Orlistat was dissolved in phosphate buffer saline before tube irrigation to mice. During the whole course, the body weight was regularly measured once a week. On day 105, the mice were euthanized after bleeding to collect plasma, and the liver was dissected, rinsed with sterilized saline and stored at −80 °C, otherwise fixed with 10% formalin for further studies. 

### 4.4. Assay for the Plasma Biochemical Parameters

The activities of GOT and GPT, and the plasma level of total cholesterol (T-CHO), low density lipoprotein-cholesterol (LDL-C), high density lipoprotein-cholesterol (HDL-C), triglyceride (TG), glucose (GLU) and uric acid (UA) were measured using the Fuji DRI-CHEM 3500 S plasma biochemistry analyzer (Fujifilm Corporation, Tokyo, Japan). 

### 4.5. Immunoassay for the Plasma Level of Insulin, Leptin, and Adiponectin 

The plasma levels of insulin, leptin, and adiponectin in mice were determined by the mouse insulin ELISA enzyme immunoassay kit (Mercodia AB, Uppsala, Sweden) according to the instruction given by the manufacturer and the optical density was read at 450 nm with the ELISA reader (VersaMax, Molecular Devices, Sunnyvale, CA, USA).

### 4.6. Western Blotting

Following our previous report [13], the expression of proteins, including Sirt1, AMPK, p-AMPK, SREBP-1c, and PPARγ in liver tissues was measured. In brief, the liver tissues were homogenized in the RIPA buffer containing protease inhibitors. The homogenate was centrifuged at 10,000× *g* for 5 min. The supernatant was separated and the protein content was determined and frozen at −80 °C until use. An aliquot of the supernatant, containing 40 μg of protein, was measured and mixed with 1/5 × Laemmli sample buffer (60 mM Tris-HCl pH 6.8, 25% glycerol, 2% SDS, 14.4 mM β-mercaptoethanol and 0.1% bromophenol blue, denatured by heating at 95 °C for 5 min). The protein samples were then separated on a 10% SDS-PAGE and electro-blotted to the nitrocellulose membranes. After blocking with TBS buffer (20 mM Tris–HCl, 150 mM NaCl, pH 7.4) containing 5% non-fat milk, the membrane was incubated overnight at 4 °C with various specific antibodies including AMPK (1:1000; # ab1315120), p-AMPK (1:1000; #ab23875), PPARγ (1:500; #ab45036) and SREBP-1 (1:5000; # ab26481) from Abcam (Cambridge, UK), Sirt1 (1: 1000; #131611AP) from Proteintech Group Inc. (Rosemont, USA) and β-actin (1:3000; #MAB1501; Millipore, Billerica, MA, USA), followed by treatment with horseradish peroxidase-conjugated anti-mouse IgG. The results were visualized with the ECL chemiluminescent detection kit (PerkinElmer, Waltham, MA, USA) and quantified by with the Image J gel analysis software.

### 4.7. Histological Examination of the Hepatic Tissues

The hematoxylin-eosin staining was applied to the histological examination of mice livers. Tissues were formalin-fixed, embedded in paraffin, 2 μm sectioned, and subjected to H&E by conventional protocol, and the images were photographed. 

### 4.8. Statistical Analysis

Data are expressed as mean ± standard error of mean (SEM) and analyzed using one-way ANOVA, followed by the Least Significant Difference (LSD), for comparing the inter-group variation of means. All analyses were statistically treated with SPSS statistics for Windows, version 22.0. Unless specified otherwise, a *p*-value < 0.05 is considered significant.

## 5. Conclusions

Dietary supplementation from fermentation products may be used as strategies for preventing or alleviating the fatty liver symptoms. The fungi polysaccharide, Antrodan, a β-glucan isolated from *A. cinnamomea* mycelia, has shown its beneficial effects against NAFLD development by suppressing plasma MDA, GOT, GPT, total cholesterol, triglycerides, glucose, insulin, upregulating adiponectin, leptin, pAMPK, Sirt1, and downregulating PPARγ and SCEBP-1c, which apparently have covered a complete spectrum of therapeutic benefits to alleviate the liver injuries. These results indicate the possible therapeutic potential of Antrodan in preventing or amelorating the HFD-induced NAFLD and its progression to NASH. Simple and cost-effective preparation of Antrodan that provides kilogram amount for further preclinical study in NAFLD is currently underway.

## Figures and Tables

**Figure 1 ijms-21-00360-f001:**
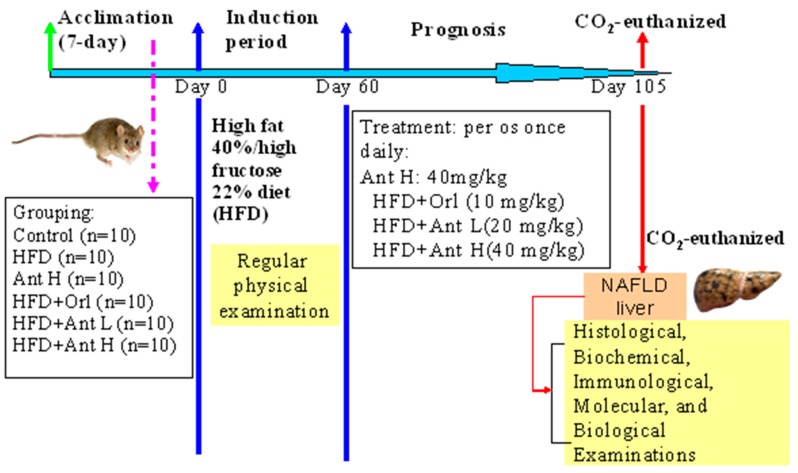
The time course of scheduled experiment to assess the alleviative effect of Antrodan for high fructose diet (HFD)–induced fatty liver in C57BL/6 mice. HFD: high fat 40% and high fructose 22% diet. Ant: Antrodan. Ant L: low dose Ant (20 mg/kg). Ant H: high dose Ant (40 mg/kg).

**Figure 2 ijms-21-00360-f002:**
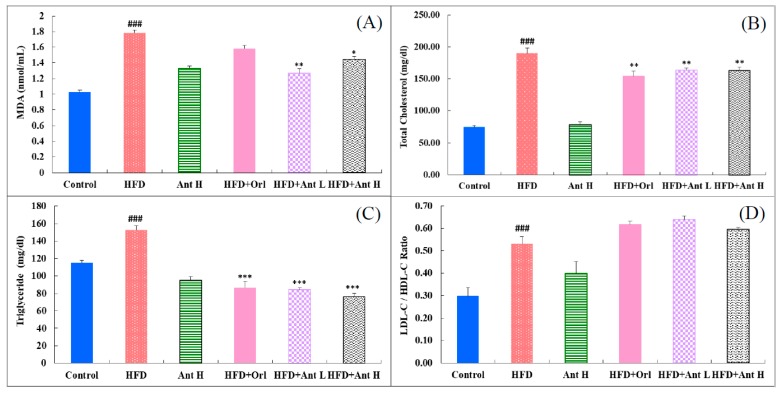
Effects of Antrodan on the plasma lipid peroxidation (**A**) and lipid profiles (**B**–**D**) in HFD-fed mice. Dada are expressed as mean ± SEM (*n* = 10). HFD: high fat 40% and high fructose 22% diet. Ant: Antrodan. HFD+Orl: HFD+Orlistat (10 mg/kg). HFD+Ant L: HFD+Ant (20 mg/kg), HFD+Ant H: HFD+Ant (40 mg/kg). ### *p* < 0.001 and ## *p* < 0.01 compared to the control; *** *p* < 0.001 and ** *p* < 0.01 compared to the HFD group.

**Figure 3 ijms-21-00360-f003:**
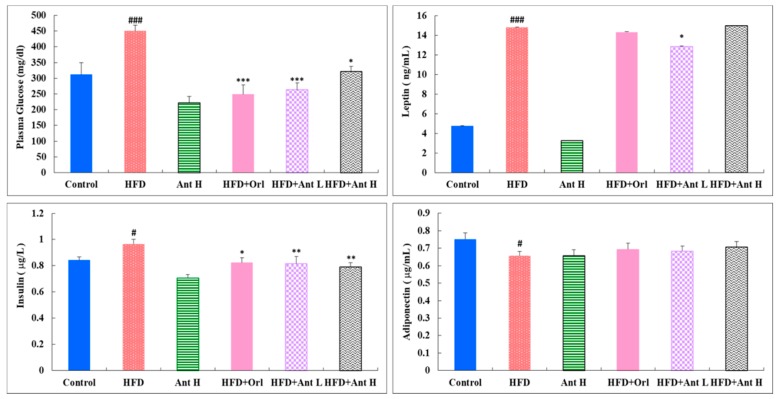
Effect of Antrodan on the plasma levels of glucose, insulin, leptin, and adiponectin in the HFD-fed mice. Data are expressed as mean ± SEM (*n* = 10). HFD: high fat 40% and high fructose 22% diet. Ant: Antrodan. Ant H: Ant 40 mg/kg, HFD+Orl: HFD+Orlistat (10 mg/kg), HFD+nt L: HFD+Ant (20 mg/kg), HFD+Ant H: HDF+Ant (40 mg/kg). ### *p* < 0.001 and # *p* < 0.05 compared to the control; ****p* < 0.001, ** *p* < 0.01 and * *p* < 0.05 compared to the HFD group.

**Figure 4 ijms-21-00360-f004:**
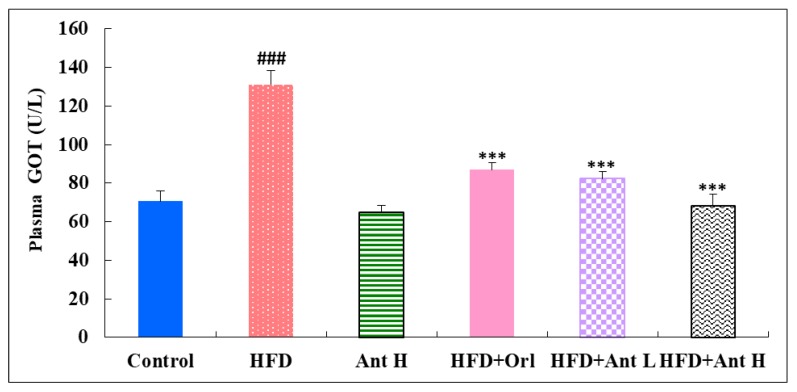
Effects of Antrodan on the plasma levels of GOT, GPT, and uric acid in HFD-fed mice. Values are expressed as the mean ± SEM (*n* =10). ### *p* < 0.001, ## *p* < 0.01 and # *p* < 0.05 compared to the control; *** *p* < 0.001, ** *p* < 0.0.01 and * *p* < 0.05 compared to the HFD group.

**Figure 5 ijms-21-00360-f005:**
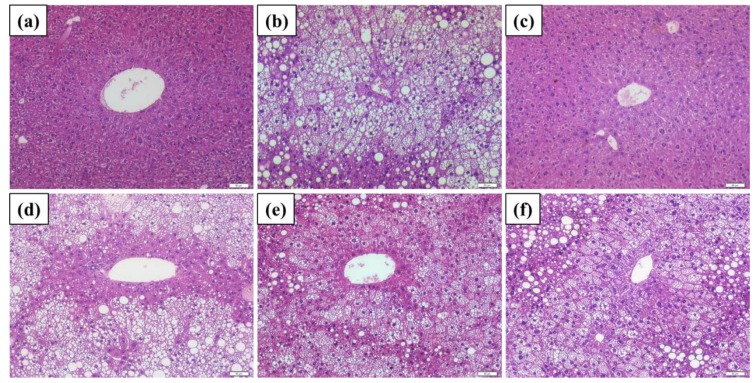
Liver biopsy with Hematoxylin and Eosin staining. HFD: high fat 40% and high fructose 22% diet. Ant: Antrodan; (**a**) control; (**b**) steatosis caused by HFD; (**c**) HFD+Ant-L (20 mg/kg). (**d**) HFD+Orl: HFD+orlistat (10 mg/kg). (**e**) HFD+Ant-L (20 mg/kg), and (**f**) HFD+Antr (40 mg/kg). Scale bar: 100 μm. (Magnification, 200×).

**Figure 6 ijms-21-00360-f006:**
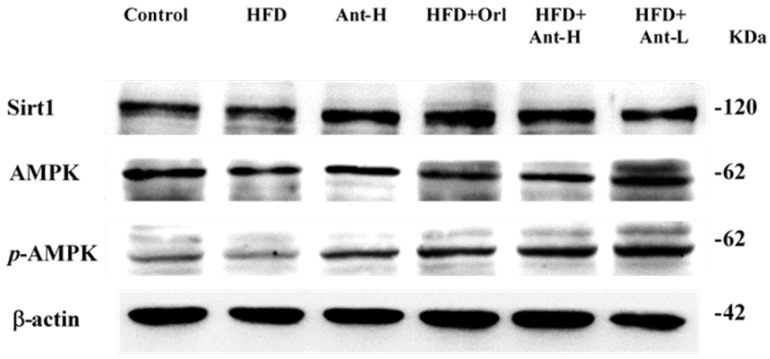
Effect of Antrodan on high-fat/high-fructose diet-induced expression of Sirt1, AMPK and p-AMPK in the liver tissues of mice. Data are expressed as the mean ± SEM (*n* = 10). # *p* < 0.05 and ## *p* < 0.01 compared with the control. * *p* < 0.05, ** *p* < 0.01 and *** *p* < 0.001 compared with the HFD group.

**Figure 7 ijms-21-00360-f007:**
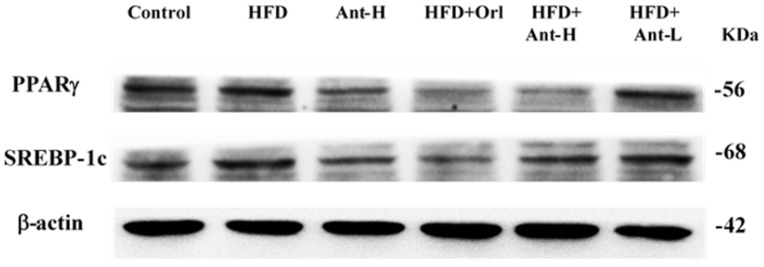
Effect of Antrodan on high-fat/high-fructose diet-induced expression of PPARγ and SREBP-1c in the liver tissues of mice. Data are expressed as the mean ± SEM (*n* = 10). # *p* < 0.05 and ## *p* < 0.01 compared with the control. * *p* < 0.05 and ** *p* < 0.01 compared with the HFD group.

**Figure 8 ijms-21-00360-f008:**
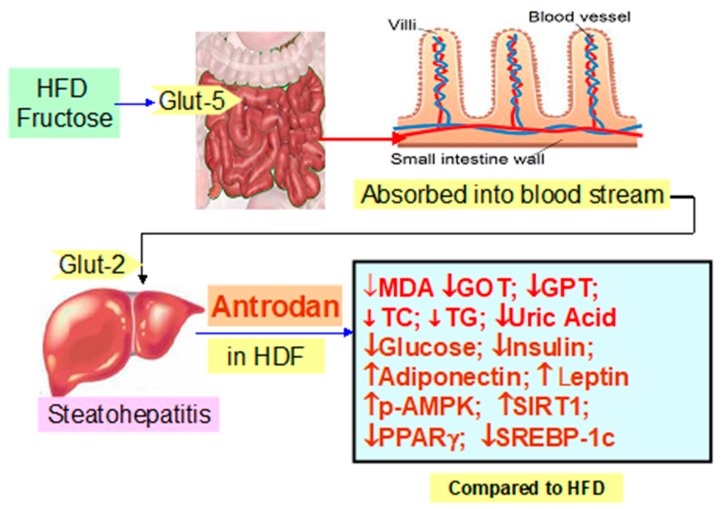
Summary of the results in this study. GOT: Glutamate-oxaloacetate transaminase; GPT: Glutamate-pyruvate transaminase; p-AMPK: phosphor-AMP-activated protein kinase; PPARγ: Peroxisome proliferator-activated receptor gamma; SREBP-1c: Sterol regulatory element-binding protein-1c. Fructose contained in the high fat HFD is transported via glucose transporter 5 (Glut-5) located on the villi in intestine into the blood stream and then distributed to many tissues, in particular, the liver carrying Glut-2 via which fructose is transferred into the cytoplasm of cells, where HFD induces steatohepatitis, which can be alleviated by Antrodan at 20–40 mg/kg.

**Figure 9 ijms-21-00360-f009:**
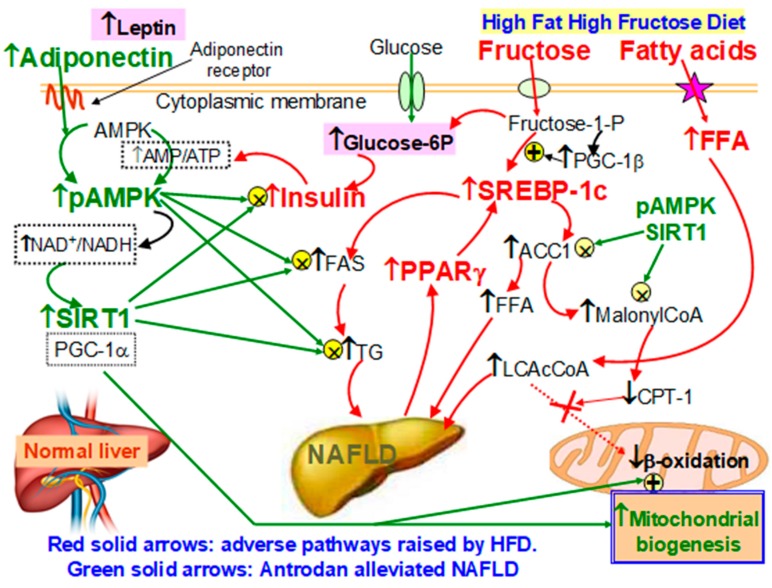
Antrodan alleviated the HFD-induced NAFLD via the. AMPK/SREBP-1c/PPARγ pathway HFD: High fat-high fructose diet. NAFLD: non-alcoholic fatty liver disease. ACC1: Acetyl-CoA carboxylase-1; AMPK: AMP-activated protein kinase; p-AMPK: phosphor-AMP-activated protein kinase; FAS: fatty acid synthase; FFA: free fatty acids; LCAc-CoA: long chain acyl CoA. PPARγ: Peroxisome proliferator-activated receptor gamma; SREBP-1c: Sterol regulatory element-binding protein-1c. TG: triglycerides; CPT-1: carnitine palmitoyltransferase I. The major part of fructose was transported into the hepatocytes is converted into glucose, which induces insulin resistance. The latter in turn increases the low energy index AMP/ATP, together with the adiponectin induced by Antrodan induces the conversion of AMPK into pAMPK. pAMPK increases the ratio NAD+/NADH and upregulates SIRT1. Working together with pAMPK, SIRT1 suppressed the insulin resistance and the level of TG, and the activity of FAS, ACC1 and malonyl-CoA biosynthesis, leading to enhanced CPT-1 and β-oxidation.

**Table 1 ijms-21-00360-t001:** Effects of Antrodan and Orlistat on body weight and liver weight in a high-fat/high-fructose diet (HFD)-fed mice model.

Group	Control	HFD	Ant-H	HFD+Orl	HFD+Ant-L	HDF+Ant-H
Body weight (g)	25.47 ± 0.58	34.20 ± 1.11 ^###^	26.50 ± 0.63	33.80 ± 0.92 **	32.40 ± 0.89	31.98 ± 0.56 *
Liver weight (g)	1.15 ± 0.04	2.05 ± 0.21 ^###^	1.08 ± 0.07	1.89 ± 0.14	1.87 ± 0.15	1.61 ± 0.05 **
Liver weight/Body weight (%)	4.49 ± 0.14	5.93 ± 0.43 ^#^	4.04 ± 0.19	5.58 ± 0.35	5.71 ± 0.32	5.04 ± 0.09 *

HDF: high-fat/high-fructose diet. Ant: Antrodan. Orl: Orlistat. Ant H: Ant 40 mg/kg; HFD+Orl: HDF+Orlistat (10 mg/kg); HFD+Ant L: HDF+Ant (20 mg/kg), HFD+Ant H: HFD+Ant (40 mg/kg). One way ANOVA is followed by the post-hoc LSD test. Values are expressed as the mean ± SEM (*n* = 10); ^#^
*p* < 0.05 and ^###^
*p* < 0.001 compared to the control; * *p* < 0.05, ** *p* < 0.01 compared to the HFD group.

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
