# Peer review of "Antrodan Alleviates High-Fat and High-Fructose Diet-Induced Fatty Liver Disease in C57BL/6 Mice Model via AMPK/Sirt1/SREBP-1c/PPARγ Pathway"

_ijms, 2020, doi:10.3390/ijms21010360_

Round 1
Reviewer 1 Report
In this paper, Chyau and collaborators analyze the effect and the mechanism of Antrodan in reducing NAFLD in an experimental model obtained by the administration of HFD and fructose (HFHF diet) to mice.
Although the paper has merit, in particular due to the dramatic relevance of the topic (the clinical management of NAFLD and NASH, which are at the moment diseases without approved therapeutic options), some improvement are needed before publication.
First of all, I would recommend a check of the language by an English mother tongue.
Major points:
the authors did not discuss about the insulin resistance of these animals and the possible effect of the antrodan and the other substances they tested on this parameter. For instance, performing a glucose tolerance test would offer additional information about this issue. liver function: how about the other marker of liver function (apart from GOT and GPT)? Are transaminases, ALP, albumin and bilirubin influenced by the diet and/or the treatments? conclusion: the conclusion is focused on the clinical relevance, but the study is preclinical! please re-formulate the conclusion, stressing a possible translational relevance.Minor points:
many typos in the text the figure in the supplementary in my opinion can be removed western blot analysis: the molecular weight of the proteins should be indicated in the figures.Author Response
Enclosed please find our revised manuscript entitled " Antrodan Alleviates High-fat and High-fructose Diet-induced Fatty Liver Disease in C57BL/6 Mice Model via AMPK/Sirt1/SREBP-1c/PPARg Pathway", which has been revised in according to the reviewers’ comments.
All of the revised text has been highlighted in red with the "Track Changes” function according to your guidance.
We have answered the questions from the two reviewers’ comments item-by-item.
The response has been submitted and attached as a word file.

Reviewer 2 Report
Overall the study is well designed and provides useful information
Author Response
Reviewer 2
Overall the study is well designed and provides useful information.
Ans. Thank you very much for your kind review and for your valuable suggestions.
Round 2
Reviewer 1 Report
The authors have significantly improved the quality of their manuscript, which is now suitable for publication.